# Cyclodextrins: Enhancing Drug Delivery, Solubility and Bioavailability for Modern Therapeutics

**DOI:** 10.3390/pharmaceutics17030288

**Published:** 2025-02-22

**Authors:** Oana Elena Nicolaescu, Ionela Belu, Andreea Gabriela Mocanu, Valentin Costel Manda, Gabriela Rău, Andreea Silvia Pîrvu, Cătălina Ionescu, Felicia Ciulu-Costinescu, Mariana Popescu, Maria Viorica Ciocîlteu

**Affiliations:** 1Department of Pharmaceutical Technique, Faculty of Pharmacy, University of Medicine and Pharmacy of Craiova, 2 Petru Rareş Street, 200349 Craiova, Dolj County, Romania; oana.nicolaescu@umfcv.ro (O.E.N.); ionela.belu@umfcv.ro (I.B.); 2Department of Instrumental and Analytical Chemistry, Faculty of Pharmacy, University of Medicine and Pharmacy of Craiova, 2 Petru Rareş Street, 200349 Craiova, Dolj County, Romania; valentin.manda@umfcv.ro (V.C.M.); maria.ciocilteu@umfcv.ro (M.V.C.); 3Department of Organic Chemistry, Faculty of Pharmacy, University of Medicine and Pharmacy of Craiova, 2 Petru Rareş Street, 200349 Craiova, Dolj County, Romania; gabriela.rau@umfcv.ro; 4Department of Biochemistry, Faculty of Medicine, University of Medicine and Pharmacy of Craiova, 2 Petru Rareş Street, 200349 Craiova, Dolj County, Romania; andreea.pirvu@umfcv.ro; 5Department of Chemistry, Faculty of Sciences, University of Craiova, 107i Calea București Street, 200512 Craiova, Dolj County, Romania; catalina.ionescu@edu.ucv.ro; 6Department of Pharmacy, University Titu Maiorescu, 16 Gheorghe Șincai Street, 031593 București, Romania; felicia.ciulu-costinescu@prof.utm.ro; 7Department of General and Inorganic Chemistry, Faculty of Pharmacy, University of Medicine and Pharmacy of Craiova, 2 Petru Rareş Street, 200349 Craiova, Dolj County, Romania; mariana.popescu@umfcv.ro

**Keywords:** cyclodextrins, drug delivery systems, inclusion complexes, solubility enhancement, bioavailability, pharmaceutical formulations

## Abstract

Cyclodextrins (CDs) have revolutionized the pharmaceutical industry with their ability to enhance the stability, solubility, and bioavailability of a wide range of active substances. These cyclic oligosaccharides, with a unique hydrophilic exterior and hydrophobic cavity, form inclusion complexes with poorly soluble drugs, improving their pharmacokinetic profiles and therapeutic efficacy. This review explores the multifaceted roles of cyclodextrins in pharmaceutical formulations, ranging from oral, ophthalmic, parenteral, and topical applications to their emerging use in targeted therapies, gene delivery, and treatment of neurodegenerative, cardiovascular, and infectious diseases. Cyclodextrins not only improve drug solubility and controlled release but also reduce toxicity and side effects, leading to safer and more effective treatments. Recent advancements, such as cyclodextrin-based nanoparticles, offer promising pathways for cancer therapy, chronic disease management, and personalized medicine. As research continues, cyclodextrins remain at the forefront of innovation in drug delivery systems, ensuring better patient outcomes and expanding the possibilities of modern therapeutics.

## 1. Introduction

Cyclodextrins have currently become extremely important in the pharmaceutical industry due to their versatility in improving the stability, solubility and bioavailability of various active substances [1]. Cyclodextrins are a class of cyclic oligosaccharides containing α-1,4-linked glucose units which are obtained through the enzymatic conversion of starch. Cyclodextrins contain oligosaccharides with a specific structure namely a hydrophilic outer surface and a hydrophobic interior [2,3]. This allows them to form inclusion complexes with several hydrophobic molecules (Figure 1), resulting in an improved solubility and stability [3]. Poor water solubility is a major concern in drug formulation as it is estimated that approximately 40% of the approved drugs and 90% of the drugs in development may fall in this category [3]. Therefore, it is possible to use cyclodextrins in various fields such as the pharmaceutical and cosmetic industry, biotechnology, nanotechnology or medicine [2].

The size of the cavity plays an important role in establishing which cyclodextrin is selected (Figure 1). While α and γ cyclodextrins contain either a smaller or larger cavity diameter compared to the drugs ready for inclusion, β cyclodextrins allow the inclusion of a number of drugs [2].

The research in this field has remained constant in the last 10 years, as the number of publications on PubMed, Figure 2, proves.

A number of studies have highlighted the applications of cyclodextrins in pharmaceutical formulations, including better therapeutic efficiency of drugs, reducing toxicity, and increasing the stability of sensitive compounds [3,4]. Studies have demonstrated the benefits of cyclodextrins in increasing the solubility of hydrophobic drugs, and also recent research has extended these applications to controlled drug release and use in nanotechnology. Several studies have also attempted the addition of a third component such as polymers, hyaluronic acid, or amino acids in order to achieve better solubility and stability, targeted delivery or controlled release [3,5].

Despite significant progress, the use of cyclodextrins continues to be an active research topic, particularly in terms of optimizing molecular interactions and developing new drug delivery systems. This review aims to review and highlight the latest findings on the role of cyclodextrins in the pharmaceutical industry, highlighting innovations in the design of inclusion complexes and their impact on the bioavailability and therapeutic efficacy of drugs.

Our review may suggest new directions for exploring the extended applicability of cyclodextrin-based formulations in improving the solubility and bioavailability of other poorly soluble drugs. The development of these systems could facilitate the use of promising pharmaceutical compounds that have traditionally been limited by their poor pharmacokinetic profile.

## 2. The Effect of Cyclodextrins over the Encapsulated Active Substance

### 2.1. Changes in Physical-Chemical Properties

A number of active substances are chemically unstable or sensitive to light, heat or oxidation. They may be protected by using cyclodextrins, resulting in inclusion complexes.

An analyzed study shows the practical importance of combining nicotinic acid and ascorbic acid with cyclodextrins. This association may be successfully used in pharmaceutical practice because encapsulation inside the cyclodextrin protects them from external factors such as oxidation or structural changes [3].

Furthermore, a study found that β cyclodextrin (β-CD) greatly increases the solubility of drugs in combinations comprising acidic non-steroidal anti-inflammatory drugs and basic H2 blockers such as indomethacin-imidazole, histidine-imidazole where the solubility of the drug is enhanced by the interaction between the drugs. β cyclodextrin also increases the solubility of drugs in combinations where solubility is reduced by different interactions between the drugs such as diclofenac and famotidine [4].

A different study aimed to increase both the water solubility of vitamin A palmitate and its stability against various external factors. The stability of vitamin A palmitate complexes affected by temperature, oxygen and UV light was investigated. The results showed an increased water solubility and stability of the encapsulated vitamin A palmitate. The surface activity of the complex recommends it as a stabilizer in emulsion type formulations (Figure 3) [6].

ITH12674 is a melatonin–sulforaphane hybrid used in the treatment of ischemic stroke. It has low bioavailability due to poor water solubility and low stability. Including the drug in 2-hydroxypropyl-β-cyclodextrin (HP-β-CD) determined an increase in solubility and better stability at different pH values as well as high temperatures [7].

A reduction in photosensitivity was observed for both indomethacin [8] and procaine [9] as well as a decrease in the thermal decomposition for nitroglycerin [10] and vitamin D [11]. Furthermore, a decrease in side effects such as gastric ulcer was shown for indomethacin. Moreover, nitroglycerin complexes exhibited a better volatility and an improved sublingual efficiency.

### 2.2. Change in Solubility

A significant use of cyclodextrins is their ability to increase the water solubility of poorly soluble active substances (Table 1). This is important because a number of drugs have low water solubility which limits their bioavailability. Cyclodextrins may include hydrophobic molecules into their internal cavity, increasing the solubility and thus the oral bioavailability of these drugs [12]. Formulation of carbamazepine, a drug used to treat epilepsy, with β-CD and HP-β-CD yielded inclusion complexes with improved solubility and bioavailability. Furthermore, preclinical pharmacokinetic studies showed better C_max_, T_max_ and AUC [13].

#### 2.2.1. Mechanism of Action of Cyclodextrins in Improving Solubility

Cyclodextrins have a ring-shaped structure, with a hydrophobic inner cavity and a hydrophilic outer surface. This unique structure allows them to incorporate lipophilic (hydrophobic) molecules into their inner cavity, forming the so-called inclusion complexes. Thus, pharmaceutical substances that normally have a low water solubility become more soluble due to the fact that they are “hidden” inside the cyclodextrin cavity, which is compatible with an aqueous solution [26].

Firstly, the mechanism of cyclodextrin complex formation involves the interaction between cyclodextrin and the active substance. Cyclodextrin molecules form bonds with a lipophilic molecule through physical interactions such as van der Waals forces [27]. Then, the lipophilic part of the active substance enters the inner cavity of the cyclodextrin, which allows a more stable interaction and prevents crystallization or aggregation of the substance. This represents the formation of the inclusion complex and the result is an increase in solubility. The active substance, as a complex, is surrounded by the hydrophilic exterior of cyclodextrin, which makes it more soluble in aqueous solutions such as biological fluids (blood, intestinal fluids).

#### 2.2.2. Impact on Bioavailability

Bioavailability refers to the percentage of an active substance that reaches the systemic circulation after being administered (usually orally) and is able to produce its therapeutic effect. An active substance must be efficiently absorbed in order to reach the bloodstream, thus solubility is essential for this absorption [28].

There are several mechanisms through which cyclodextrins improve bioavailability.

Firstly, cyclodextrins improve bioavailability by improving absorption. Increased solubility in biological environments (such as stomach or small intestine) allows a faster and more efficient absorption of the active substance. In many cases, poorly soluble molecules tend to remain in the digestive tract and be eliminated before being absorbed, but cyclodextrins prevent this process [28]. Piroxicam is a non-steroidal anti-inflammatory drug with a water solubility dependent on the pH of the environment. Upon complexation with β-cyclodextrin the drug exhibited better pharmacokinetics due to a faster absorption rate from the gastrointestinal tract [29].

Secondly, they protect the active substance from degradation. In many cases, poorly soluble drugs may be degraded before absorption (e.g., by enzymes or gastric pH). Cyclodextrins can shield the active molecules from these processes, maintaining their stability up until absorption. Furthermore, cyclodextrins ensure a better distribution. Famotidine was associated with 2-hydroxypropyl-ß-cyclodextrin to form an inclusion complex in order to decrease the degradation rate of the drug under acidic conditions [30]. A study conducted by Wang et al. showed that complexation of resveratrol with SBE-β-CD protected the drug from degradation in biological matrices; the inclusion complexes containing more than 80% of the drug by the end of 192 h [31].

Moreover, cyclodextrin–drug complexes have a better distribution in solution which may lead to sustained and controlled release of the active substance, thus maximizing the therapeutic effect over a longer time period [32].

Lastly, they reduce inter-patient variability. By improving pharmacokinetics such as absorption and distribution, cyclodextrins can reduce inter-patient variations in bioavailability, leading to a more predictable and effective therapeutic response. Actitretin is an oral retinoid used in the treatment of psoriasis. Upon oral administration of actitretin an inter-patient variability was observed in the pharmacokinetics of the drug. However, after complexation with randomly substituted methyl β-cyclodextrin the oral kinetics in rats became less variable due to rapid and complete absorption [32].

Several active substances in various pharmaceutical forms were shown to have a better bioavailability.

Oral formulations

A number of orally administered drugs with poor water solubility were recently reformulated using cyclodextrins. Itraconazole, an antifungal, has been formulated using cyclodextrins in order to improve its bioavailability. This led to a better absorption and increased the effectiveness of the drug in the treatment of fungal infections [21].

A different study showed that inclusion complexes between itraconazole and various cyclodextrins have a significantly improved solubility. Namely, complexes obtained at a 1 to 3 molar ratio (itraconazole-cyclodextrin) were able to improve solubility and subsequently drug bioavailability without affecting its therapeutic effect [33].

Cefdinir, an antibiotic with poor water solubility was complexed with several cyclodextrins such as β-CD, HP-β-CD, and sulfobutyl ether 7-β-cyclodextrin (SBE7-β-CD), γ-cyclodextrin (γ-CD) and formulated into tablets. Complexation with HP-β-CD yielded the best results in respect to drug release and the efficacy of the drug while maintaining its antimicrobial activity [34].

Ophthalmic formulations

Cyclodextrins are used in eye preparations to dissolve and stabilize hydrophobic active substances. Dexamethasone, a corticosteroid used to treat ocular inflammation, has been formulated with cyclodextrins to improve solubility and reduce ocular irritation [35]. Loteprednol etabonate a corticosteroid with fewer adverse effects compared to dexamethasone has a poor water solubility, <0.001 mg/mL. Inclusion complexes of loteprednol and HP-β-CD and β-CD were formulated into drops, gels and ocuserts. All inclusion complexes exhibited enhanced drug solubility and release profile. Loteprednol-HP-β-CD complex showed a better solubility due to its more amorphous nature. Furthermore, ocuserts with loteprednol-HP-β-CD displayed a better drug release profile [36].

Azithromycin, an antibiotic with poor water solubility was associated with sulfobutylether-β-cyclodextrin in order to improve solubility and stability of the drug. The inclusion complex was formulated as an in situ gel. The association with cyclodextrin showed several advantages such as increase in viscosity, extended release of the drug, a reduction in degradation of the drug in the antimicrobial study [37].

Parenteral formulations

Parenteral formulations must comply with several high quality standards such as sterility, isotonicity, isohidricity, free from pyrogens and physical and chemical stability [38]. Cyclodextrins contribute to the intravenous administration of hydrophobic or lipophilic substances that are normally poorly soluble or chemically unstable. For example, amphotericin B, an antifungal with high toxicity and low solubility, was included in cyclodextrins in order to improve solubility and reduce adverse effects [39]. Propofol was associated with sulfobutyl ether-β-cyclodextrin in a parenteral formulation in order to reduce side effect such as allergic reactions and pain on injection while maintaining the pharmacokinetic and pharmacodynamic characteristics similar to the lipid formulation [40]. Furthermore, doxorubicin (DOX) was formulated with M-β-CD [41] and 6-monoamino-β-CD [42] as an inclusion complex, resulting in decreased cardiotoxicity and increased the water solubility of the drug.

Solutions for chronic diseases

Using cyclodextrins in treating chronic diseases such as hypertension or diabetes yields a constant therapeutic effect due to increased stability and controlled release of the drug [43].

### 2.3. A Reduction in Side Effects

Cyclodextrins may reduce side effects of drugs through several mechanisms, mainly due to their ability to form inclusion complexes with active molecules. This process can alter the release, absorption and distribution of the drug, providing several advantages such as stabilizing and protecting the drug and controlled release of the drug [28,29,30].

#### 2.3.1. Stabilizing and Protecting the Drug

Cyclodextrins protect drugs from premature degradation caused by environmental factors such as light or oxidation, which could reduce efficacy or increase toxicity. By protecting the drug until absorption, cyclodextrins ensure that an effective dose reaches the bloodstream without generating toxic substances during the degradation process. Inclusion complexes of naproxen, non-steroidal anti-inflammatory drug, and cyclodextrins show a better solubility and stability [44]. This allows for lower doses to be administered, thus reducing side effects. A different study showed that complexation of nicardipine with several cyclodextrins, namely HP-β-CD, β-CD, (2-hydroxyethyl)-α-cyclodextrin led to a photoprotective effect while complexation with cyclodextrin favors photodegradation [45]. Another study successfully showed that γ-CD protects ascorbic acid from oxidation in an aqueous solution as a ternary complex with poly vinyl alcohol [46].

#### 2.3.2. Controlled Release of the Drug

Controlled drug release using cyclodextrins is one of the methods used to reduce toxicity and improve therapeutic effect. This mechanism works by controlling the rate at which the drug is released into the body, thus avoiding rapid increases in plasma concentration that can lead to side effects. Basically, cyclodextrins form an inclusion complex with active drug molecules, where the drug is “trapped” in the cavity of the cyclodextrin. This complex formation helps protect the drug and gradually release it into the body in a controlled manner. A constant plasmatic concentration of the drug is maintained by slowing the release process, thus preventing concentration peaks that can lead to toxic or side effects [47,48].

This application may determine the reduction in systemic toxicity. When drugs are fast released into the body they may produce severe adverse reactions because their plasmatic concentration may reach high values in a very short time. Cyclodextrins control the release of the drug due to mucoadhesive properties or by adjusting the hydrophobicity of the guest molecule [48,49]. Furthermore, the release of the drug may be controlled by temperature, pH value, ultrasonic intensity, pressure or the degradation of the polymer [50]. This results in gradual absorption of the drug, reducing toxicity. The hydroxyl groups of CD allow structural modifications yielding hydrophobic derivatives, namely acylated and ethylated cyclodextrins, that may be used to achieve a sustained release of the drugs. Meanwhile, hydrophilic derivatives, namely SBE-β-CD and HP-β-CD, improve the solubility and bioavailability of poorly water-soluble drugs [51]. Dexamethasone, a glucocorticoid used to treat inflammatory conditions, may determine significant gastrointestinal or systemic side effects if high doses are administered. But if the drug forms an inclusion complex with a cyclodextrin, its release is controlled which reduces the risk of severe side effects [46]. Vanillin-/curcumin-β-CD inclusion complexes together with β-cyclodextrin glycosyltransferase and amyloglucosidase were formulated into a enteric-coated tablet-based controlled release system. The amount of β-cyclodextrin glycosyltransferase modulated the controlled release of the active substances in the small intestine. Moreover, the system shows great potential for the development of controlled drug delivery systems containing volatile flavor substances [50]. Formulations of nicardipine and two cyclodextrins (hydrophilic HP-β-CD and hydrophobic triacetyl-β-cyclodextrin) in different mixing ratios were obtained in order to achieve an initial fast release followed by a prolonged release. It was observed that the plasma concentration of the drug had an attenuated high peak which could translate into reduced and less frequent adverse effects of the drug [51].

Moreover, controlled release of the drug may prevent local side effects. The fast release of the drug to a targeted tissue may produce irritation or local side effects. This is also reduced through controlled release. Nifedipine, a calcium channel blocker used to treat hypertension, determines gastrointestinal irritation when fast released. Cyclodextrins control the release of the drug which reduces the risk of local irritation [52]. A nanogel eye drop formulation containing dexamethasone associated with hydroxypropyl-γ-cyclodextrin (HP-γ-CD) was shown to include 25 times more active substance compared to the commercial formulation and it provided a sustained release for more than 6 h. Furthermore, the new formulation determined no irritation or redness upon application to rabbits [49]. Moreover, controlled release helped maintain an effective drug concentration for a longer period of time, which may reduce the need for frequent drug administration and improve patient compliance [53].

Progesterone complexed with cyclodextrins formulations show a slow release of the hormone while maintaining a constant concentration and reducing the risk of sudden hormonal fluctuations that could cause adverse effects [54]. Chitosan nanoparticles loaded with naringenin–sulfobutylether–β-cyclodextrin inclusion complexes were shown to prolong the residence time and improve bioavailability in the aqueous humor which leads to reduced dosing frequency. In vivo studies determined no irritation when the formulation was applied to a rabbit’s eye [55].

### 2.4. Reducing Local Toxicity

Cyclodextrins protect tissues from direct interaction with irritating or toxic substances. They encircle the active molecule of the drug in an inclusion complex, thus protecting sensitive tissues from direct contact with the active substance. This process can reduce irritation, damage, or inflammation that may occur with some medications [56].

Cyclodextrins are used to reduce the irritation produced by ciclopirox olamine when applied locally to the skin, thus improving tolerability and reducing the risk of dermatitis or other local reactions [56]. A study conducted by Saokham et al. showed that raising CD concentration from 0 to 8% increased the amount of hydrocortisone in the donor phase. The formation of hydrocortisone–cyclodextrin inclusion complexes increased both drug partition into the skin and permeation through the skin [57].

### 2.5. Detoxifying Active Compounds

Detoxifying active compounds using cyclodextrins is an important mechanism by which these molecules can capture and neutralize toxic substances, which reduces the risk of systemic or local toxicity. Cyclodextrins have a unique, cone-shaped structure with a hydrophobic inner cavity and a hydrophilic outer layer, which allows them to incorporate hydrophobic molecules and remove them from the biological environment, protecting the body from their harmful effects [57,58,59].

The mechanism is based on the cyclodextrins forming inclusion complexes with toxic molecules or substances that can become toxic under certain conditions. In this process, the toxic molecule is “trapped” in the cyclodextrin’s hydrophobic cavity, thus preventing its interaction with tissues and cells in the body. This formation of complexes can facilitate the elimination of harmful substances from the body or reduce their absorption in the gastrointestinal system or other parts of the body [57,58,59].

Alpha-cyclodextrin can be used to bind cholesterol and reduce its levels in cardiovascular conditions related to high cholesterol. This process helps prevent the formation of atheromatous plaques in the blood vessels, thereby reducing the risk of cardiovascular diseases such as heart attack or stroke [58].

A study conducted by Jakšić concluded that sugammadex (a γ-CD derivative) and γ-CD could be used as first aid in acute intoxication with sterigmatocystin as cyclodextrins bind mycotoxins and extract them from the blood [59]. Cyclodextrins, particularly 2-hydroxypropyl-β-cyclodextrin, are used to bind cholesterol that, in Niemann–Pick disease type C, accumulates in different tissues and helps eliminate it from the cells. This may prevent damage caused by cholesterol accumulation in liver, spleen, brain and other organs [60].

In the case of ibuprofen or other non-steroidal anti-inflammatory drugs (NSAIDs) overdose, cyclodextrins can be used to reduce the absorption of these drugs, thereby preventing gastrointestinal and renal toxicity [61].

The current treatment for organophosphorus nerve agents has a limited efficiency against chemicals such as tabun and soman. Five new cyclodextrins were evaluated as a more efficient options in reducing the inhibitory effect on acetylcholinesterase and exhibited promising results in the detoxification process [62].

## 3. Pharmaceutical Forms with Cyclodextrins

Cyclodextrins are used in a wide range of pharmaceutical formulations due to their ability to improve the solubility, stability and bioavailability of active substances. Cyclodextrins can be found in various pharmaceutical forms that include tablets, capsules, powders, oral solutions and suspensions (Figure 4).

### 3.1. Tablets

Cyclodextrins may be included in a wide range of products from regular tablets to effervescent tablets, coated tablets, modified-release tablets, orally disintegrating tablets (ODTs) [63]. Cyclodextrins are included in the formulation of tablets as excipients for their ability to enhance water solubility [64], increase stability, mask the bitter taste of the drug [65,66,67], to minimize side effects and to control the release of the active substance [63].

In Nitropen, a sublingual tablet used in the treatment of angina, nitroglycerin is complexed with β-cyclodextrin. The inclusion complex displayed low volatility of nitroglycerin. Subsequently, the tablet determined no sensation of foreign body in the sublingual region and it exhibited no change in nitroglycerine content during storage time [68].

Olmetec OD tablets and olmesartan OD tablets contain olmesartan medoxomil complexed with β-CD in order to suppress the smell of the drug [69,70].

Brivaracetam is formulated with β-cyclodextrins (Brivlera), a drug used in the treatment of epilepsy, to achieve improved drug dissolution (Table 2). Furthermore, addition of cyclodextrin reduces the mixture sticking qualities during the compression process. Moreover, the compression pressure required in the tableting process is lower due to the inclusion of cyclodextrin [71].

### 3.2. Capsules

Cyclodextrins can be used for the formulation of capsules, either in powder form or as part of a mixture containing the active substance. β-CD and HP-β-CD are predominantly used in capsule formulation as shown in Table 3. Inclusion complexes increase both solubility and stability of the drug [86]. Furthermore, they improve absorption of the active substance in the gastrointestinal tract.

Omebeta contains omeprazole complexed with β-CD in order to achieve better bioavailability. Furthermore, Geng et al. showed that capsules containing omeprazole β-CD inclusion complexes displayed immediate drug release which may increase patient compliance. Complexation with β-CD ensured a high solubility and stability of the drug in the acidic environment of the stomach [87].

**Table 3 pharmaceutics-17-00288-t003:** Cyclodextrins used for the formulation of capsules.

Active Substance	Trade Name	Cyclodextrin	Effect of Cyclodextrin	Indication	Reference
4-Androstenediol	Androtest	HP-β-CD	Better water solubility of inclusion complex, increased bioavailability	Supplement	[88]
Benexate HCl	Ulgut, Lonmiel	β-CD	Increase in solubility and strong antiulcer activity	Gastritis	[89]
Curcumin extract	Curcumin Extract 45	γ-CD	Improved absorption of curcuminoids and bioavailability	Supplement	[90]
Fingolimod	Fingolimod beta	β-CD	Up to 20x increase in solubility	Multiple sclerosis	[66,91]
Omeprazole	Omebeta, Losamel, enteric capsule	β-CD	Improved bioavailability	Ulcers	[87]

### 3.3. Powders and Granules

Cyclodextrins are frequently used to formulate powders and granules, intended for either direct oral administration or reconstitution in aqueous solutions. They determine a better dispersion and increased solubility. Furthermore, they may be used to mask unpleasant taste or flavor, to avoid interactions between formulating substances or to transform liquid substances such as therapeutic oils into powder form [80,92].

Several formulations of omeprazole and cyclodextrins as powders or granules were obtained to improve both solubility and stability. Inclusion complexes of omeprazole (OME) and gamma-cyclodextrin (γ-CD) were obtained by spray-drying, coprecipitation, kneading and freeze-drying. The inclusion complexes displayed an increased release rate of the drug. Furthermore, the highest dissolution rate was achieved for the coprecipitation product [93]. Cholestyramine is used in powder form (Questran^®^) with cyclodextrins to bind bile acids and lower cholesterol. Glucagon is formulated with β-CD as Baqsimi™, a nasal dry powder formulation used in the treatment of diabetes. β-CD acts as a permeation enhancer for the drug, thus improving stability, solubility and bioavailability [81,92,94]. Differently charged β-cyclodextrins such as neutral β-cyclodextrin, anionic sulfobutylated-β-cyclodextrin sodium salt and cationic (2-hydroxy-3-*N*,*N*,*N*-trimethylamino) propyl-β-cyclodextrin chloride were formulated with meloxicam potassium as nasal powders. Cyclodextrin formulations displayed high adhesive forces. Furthermore, the anionic cyclodextrin formulation showed increased permeation of the drug [95].

### 3.4. Oral Solutions and Suspensions

Cyclodextrins are ideal for formulating solutions and suspensions by aiding with the dissolution of poorly soluble active substances in aqueous solutions [96], improving stability [97,98] and masking the unpleasant taste of drugs [46,65,67,99,100]. Thus, the oral bioavailability of the drug is improved. These characteristics recommend cyclodextrins as valuable excipients in pediatric dosage form as shown in Table 4 [101]. Midazolam can be found in oral solutions that include cyclodextrins to improve the solubility and absorption of this benzodiazepine sedative. Ibuprofen formulations use cyclodextrins such as α-cyclodextrin (α-CD), methylated-β-cyclodextrins and γ-CD to make the drug more soluble and easier to administer and to mask the taste of ibuprofen. Inclusion complexes may be formulated as oral dosage forms, namely elixirs, syrups or suspensions [102].

Itraconazole is practically water insoluble [103]. In Sporanox^®^, oral solution, itraconazole is formulated with HP-β-CD to increase both the water solubility (10 mg/mL) and bioavailability of the drug [103,104]. In a study conducted by Shehata et al. itraconazole was complexed with several cyclodextrins (α-CD, β-CD, γ-CD, HP-β-CD) to evaluate their ability in increasing the water solubility of the drug. HP-β-CD exhibited the best results [105].

**Table 4 pharmaceutics-17-00288-t004:** Cyclodextrins used for oral solutions and suspensions.

Active Substance	Trade Name	Cyclodextrin	Effect of Cyclodextrin	Indication	Reference
Itraconazole	Sporanox	HP-β-CD	Increase water solubility and bioavailability	Fungal infections	[103,104]
Larotrectinib sulfate	Vitrakvi	HP-β-CD	Solubilizing agent	Pediatric cancer	[106]
Midazolam	Ozalin	γ-CD	Increase midazolam solubility, mask bitter taste of solution	Premedication before anesthesia, in pediatric population	[107,108]
Piroxicam	Flogene	β-CD	Increased solubility, fast absorption and improved gastric tolerability	Analgesic for pediatric use	[86,109]

### 3.5. Parenteral Formulations

In parenteral formulations, cyclodextrins are used to solubilize hydrophobic drugs, allowing intravenous or intramuscular administration. Only a few cyclodextrins, namely α-CD, HP-β-CD, and SBE-β-CD are approved for parenteral administration but they are used in numerous products available on the market [64]. Other α-CD and β-CD derivatives should not be used in parenteral formulations due to nephrotoxicity and hemolytic activity [110,111]. In some cases, parenteral formulations are also used as lyophilized formulations, reconstituted before administration such as Kyprolis^®^. Carfilzomib is formulated with SBE-β-CD as an inclusion complex in Kyprolis^®^, a powder for solution for infusion, in order to improve aqueous solubility [112].

Aripiprazole forms an inclusion complex with SBE-β-CD (Abilify) in order to achieve an increased water solubility. Furthermore, the inclusion complex formulation displayed reduced tissue irritation at the injection site whereas an aripiprazole formulation was found to produce moderate to severe irritation [113].

Fosphenytoin sodium, a drug used in the treatment of epilepsy, was complexed with SBE-β-CD in order to achieve improved stability and solubility at a pH closer to physiological pH, therefore reducing stimulation (Table 5). Moreover, the inclusion complex formulation displays an increased shelf life at room temperature [114,115].

**Table 5 pharmaceutics-17-00288-t005:** Cyclodextrins used for parenteral formulations.

Active Substance	Trade Name and Formulation	Cyclodextrin	Effect of Cyclodextrin	Indication	Reference
Amiodarone	Nexterone intravenous solution	SBE-β-CD	Solubilizing agent. The inclusion complex solution is chemically stable upon storage time	Treatment and prophylaxis of frequently recurring ventricular fibrillation	[116]
Aripiprazole	Abilify, solution for injection (intramuscular)	SBE-β-CD	Increased water solubility	Schizophrenia	[69,113]
Diclofenac	Dyloject^®^ (intramuscular and intravenous use)	HP-β-CD	Increase the solubility of the drug, resulting in less clinical thrombophlebitis	Rheumatic and non-rheumatic pain	[117,118]
Diclofenac	Akis subcutaneous and intramuscular use)	HP-β-CD	Good stability, solubility, and bioavailability of the preparations	Rheumatic and non-rheumatic pain	[117,118]
Fosphenytoin	Sesquient, Cerebyx	SBE-β-CD	Improved stability and solubility	Epilepsy	[114,115]
PGE1 (Alprostadil)	Alprostadil Alphadex, Caverject intracavernous injection	α-CD	Increased stability	Buerger disease, male erectile dysfunction	[66,81,119]
Posaconazole	Noxafil	SBE-β-CD	Improved solubility and stability	Anti-infective applications	[120]
Progesterone	Lubion^®^ solution for intramuscular and subcutaneous injection	HP-β-CD	Increase in the solubility of progesterone, resulting in improved bioavailability	Luteal phase support during assisted reproductive technology	[118]
Remdesivir	Veklury, powder for solution for infusion	SBE-β-CD	Increased solubility, stability, bioavailability	COVID-19	[81,121,122]
Voriconazole	Vfend, powder for solution for infusion	SBE-β-CD	Increased solubility, better safety profile (no deterioration of renal function) in patients with renal failure	Fungal infections	[123,124]
Ziprasidone mesylate	Geodon, powder for solution for injection	SBE-β-CD	Solubilizing agent	Schizophrenia	[125]

### 3.6. Topical Formulations

Cyclodextrins are used in topical formulations such as gels and creams to stabilize active ingredients, enhance permeation for topical formulations, and to control/modify their release (Table 6). Furthermore, they can decrease the in vitro release rate of corticosteroids from lipophilic ointment bases and enhance drug release from hydrophilic ointment bases [81].

Some authors suggest that cyclodextrins enhance permeation through biomembranes such as eye cornea or skin by extracting lipophilic components from the membranes (Table 7) [126,127]. To the contrary, several studies showed that a cyclodextrin excess in an aqueous vehicle determines decreased penetration of lipophilic drugs through membrane [126,128,129]. Cyclodextrins yield inclusion complexes with lipophilic drugs that release the drug to the surface of a biological membrane namely, mucosa, skin or cornea [130]. Furthermore, tear liquid dilutes the inclusion complex and promotes the release of the drug. A biological membrane such as the eye cornea has low affinity towards inclusion complexes or large cyclodextrin molecules [126,131].

Cyclodextrins may also be included in sunscreens as they protect UV filters and antioxidants from degradation determined by sunlight and oxygen exposure, therefore improving stability. Moreover, they modulate the release of antioxidants and promote penetration through skin, thus increasing bioavailability [132,133]. When included in sunscreens, cyclodextrins can boost the ultraviolet radiation filtering [134,135].

Metronidazole forms an inclusion complex with β-CD which acts as a solubility enhancer (Table 6). Niacinamides, a second solubility enhancer, displays synergistic effect in combination with cyclodextrin, allowing for a lower cyclodextrin concentration. This reduced the formulation and preparation cost. Furthermore, the formulation does not contain high concentration of organic solvents that exhibit an irritating effect on skin, which allows applications in dermatological conditions, namely rosacea [135].

**Table 6 pharmaceutics-17-00288-t006:** Cyclodextrins in topical pharmaceutical forms for dermal drug delivery.

Active Substance	Trade Name	Cyclodextrin	Effect of Cyclodextrin	Indication	Reference
Benzoyl peroxide	Nujevi Acne	γ-CD	Decrease drug degradation	Acne	[132]
Dexamethasone	Glymesason, ointment	β-CD	Increased solubility up to 33 fold in a 1:1 complex	Dermatitis	[66,81]
Metronidazole	Metrogel, gel	β-CD	Solubility enhancer	Rosacea	[135]
Salicylic acid	Age Defying Blemish Treatment, Lipo™ CD-SA	HP-β-CD	Reduces skin irritation and stinging	Skin care	[136]

**Table 7 pharmaceutics-17-00288-t007:** Cyclodextrins in topical pharmaceutical forms for ophthalmic drug delivery.

Active Substance	Trade Name	Cyclodextrin	Effect of Cyclodextrin over the Active Substance	Indication	Reference
Chloramphenicol	Clorocil	Methyl-β-cyclodextrin	Increased stability	Eye infections	[137,138]
Diclofenac	Voltaren Ophtha Eye Drops	HP-γ-CD	Drug solubility enhancer and penetration promoter, reduction in ocular toxicity	Pain and inflammation after ocular surgery and seasonal allergic conjunctivitis	[69,139]
Lanosterol	Lanomax	HP-β-CD	Increased water solubility, improved bioavailability	Cataract therapy for dogs	[140]
Levocabastine	Allergiflash	HP-β-CD	Solubilizing agent, increases bioavailability	Allergic conjunctivitis	[141]
Olopatadine	Opatanol	HP-β-CD	Increases stability by preventing precipitation or crystallization	Allergic conjunctivitis	[142]

### 3.7. Suppositories

The rectal route of administration has several disadvantages including small volume of rectal fluid for drug dissolution and a limited area for absorption [71,143]. In suppositories, cyclodextrins can help solubilize and release the active ingredients into the rectum, increasing their absorption and subsequently their bioavailability [81,142,144]. Furthermore, they may improve chemical stability and promote permeation through the rectal mucosa [81,142]. They also reduce local irritation, resulting in increased patient compliance [142]. Studies showed that high quantities of β-CD and as much as 12% HP-β-CD may be used in suppository formulation as they do not produce damage of the rectal epithelium [145].

Prostaglandin urethral suppositories (Muse^®^) for the treatment of erectile dysfunction use alpha cyclodextrins to increase stability and release of alprostadil. Diclofenac can be formulated with cyclodextrins as suppositories to improve release and absorption through the rectal mucosa [146].

Mobitil contains inclusion complexes of meloxicam and β-CD formulated as suppositories. A study conducted by Gowthamarajan showed an improved release of meloxicam from suppositories containing the inclusion complex [147].

### 3.8. Aerosols

In products intended for pulmonary administration, cyclodextrins help to solubilize the active substances and to attain an efficient dispersion in the respiratory tract. Furthermore, studies show that inclusion complexes improve the aerosolization properties of the pharmaceutical form and drug solubility in lung fluids [81,148].

Sulfoalkyl ether gamma-cyclodextrin (SAE-γ-CD) is used in inhalation solutions with budesonide, a corticosteroid, to improve the stability and solubility of the drug. Furthermore, the formulation displays several advantages such as reduced toxicity and treatment, improved drug delivery time and ease of manufacture [149]. In inhalers, cyclodextrins help solubilize and stabilize formoterol.

### 3.9. Transdermal Patches

Transdermal formulations must ensure that the drug penetrates the skin and reaches systemic circulation [81]. Cyclodextrins can be included in transdermal patches to improve the controlled release and skin penetration of active ingredients. Moreover, they improve drug solubility and reduce local inflammation [86].

Smoking cessation patches contain cyclodextrins such as β-CD to improve the controlled release of nicotine through the skin [150]. Furthermore, patches containing inclusion complexes formulated with nicotine and either β-CD or M-β-CD showed increased stability and skin permeation of the drug [151].

Rivastigmine patches contain cyclodextrins to improve penetration of the drug through skin and ensure gradual release in the treatment of Alzheimer’s disease.

Cyclodextrins such as HP-β-CD were used to increase antiemetic drug solubility in ready to use transdermal vehicles such as Pentravan^®^ or Lipovan^®^ in order to promote transdermal drug delivery. This type of formulation can be easily obtained in the pharmacy of a hospital [152].

Thus, cyclodextrins are versatile molecules that improve physical-chemical properties and bioavailability for numerous active substances in many different pharmaceutical forms.

Research conducted in the field of cyclodextrin formulations show great results in improving drug delivery and efficiency [66].

### 3.10. Nanotechnology-Based Drug Delivery Systems

Cyclodextrins are used to develop nanoparticles that improve the solubility and bioavailability of sparingly soluble drugs, such as paclitaxel and camptothecin, used in oncology. These new formulations provide targeted delivery and reduce side effects, improving therapeutic efficacy [66].

Cyclodextrin along with polyethylene glycol were used to increase roxithromycin loading capacity into superparamagnetic nanoparticles. The targeted drug delivery systems showed antimicrobial activity against *E. coli* and *S. aureus* [153]. Thyme essential oil was encapsulated into β-cyclodextrin nanosponges, resulting in an increased antibacterial activity. Furthermore, the water solubility of the poorly soluble essential oil was increased 15 fold [154].

Cyclodextrins are being researched as one of the components in gene delivery systems. They display several advantages such as reducing immunogenicity, incorporating small drug molecules and act as linkers [155]. Cyclodextrins are used in combination with other excipients to deliver nucleic acids such as interfering RNA (siRNA). These systems aim a controlled release and improved cellular uptake, showing promise for future clinical applications [66]. The hydroxyl groups on the outside of β-CD or γ-CD were crosslinked after activation with the amino groups found on polyethyleneimine, leading to lower cytotoxicity and improved gene transfer [155]. A β-CD polymer nanocarrier for the delivery of RNA-cleaving DZ exhibited a better inhibitory effect of cell proliferation in the presence of doxorubicin compared to the drug alone [118]. Another study succeeded in the targeted delivery of DNAzymes via chitosan/β-cyclodextrin complexes for restoring chemosensitivity in the doxorubicin-resistant breast cancer cell line [156].

Studies show that cyclodextrin derivatives may be used to incorporate antiviral agents, improving their stability and efficacy in treating viral infections, including COVID-19 [157]. Furthermore, cyclodextrins act as cryopreserving agents and adjuvants in vaccine formulation [158]. Veklury, a powder for solution for infusion is the first antiviral formulation on the market. It includes remdesivir complexed with SBE-β-CD in order to increase the stability and solubility of the drug [81,121,122]. Boudad et al. incorporated a saquinavir HP-β-CD inclusion complex into poly (alkyl cyanoacrylate) nanoparticles. HP-β-CD increased saquinavir water solubility by 400 fold and nanoparticle loading by 20 fold, thus showing great promise as an oral formulation in the treatment of HIV [159]. Janssen vaccine against SARS-CoV2 infection marks the first approved use of cyclodextrins, namely HPβCD, as a cryopreservative in this field [160]. Porcine Circovirus Vaccine Suvacyn PCV contains an inactivated porcine circovirus recombinant virus (CPCV) 1-2 as the active substance and both squalene and SL-β-CD as adjuvants. The cyclodextrin produced a better immune response and a fast onset of immunity compared to the previous formulation [160].

These new formulation underline the versatility of cyclodextrins in modern therapy from improving drug solubility to targeted delivery in oncology, gene and antiviral therapies.

## 4. The Use of Cyclodextrins in Various Diseases

Cyclodextrins are currently used in treating various diseases due to their significant role in increasing drug solubility and stability and targeted release. There are a number of therapeutic areas that use cyclodextrins.

### 4.1. Neurodegenerative Disorders

Cyclodextrin, particularly HP-β-CD, is researched for the treatment of diseases such as Alzheimer’s and Parkinson’s diseases. These conditions are related to cholesterol metabolism. Both β-CD and M-β-CD exhibited the highest affinity for cholesterol followed by HP-β-CD and heptakis-2,3,6-tris-*O*-methyl β-CD (TRIMEB) [161]. M-β-CD is only used in topical formulations due to its renal toxicity and hemolytic character, thus making HP-β-CD a far better candidate [147]. HP-β-CD helps regulate cholesterol and it is well tolerated, exhibiting therapeutic potential [162]. Cyclodextrins have received significant interest in treating neurodegenerative diseases due to their ability to improve drug delivery, facilitate cellular detoxification, and influence lipid metabolism. Due to these properties cyclodextrins may be useful in conditions such as Alzheimer’s disease, Parkinson’s disease, and Niemann–Pick type C disease, which involve the accumulation of toxic substances or a flaw in the elimination of certain compounds from the brain.

#### 4.1.1. Improved Drug Delivery to the Brain

One of the major challenges in the treatment of neurodegenerative disorders is the brain blood permeability which limits the delivery of drugs into the brain. Cyclodextrins may help improve drug transport through the membrane by forming inclusion complexes with active molecules, increasing bioavailability in the central nervous system. Studies display the use of cyclodextrins to facilitate the delivery of anti-inflammatories or antioxidants that may help protect neurons in neurodegenerative diseases. Newer β-CD variations are being studied in order to achieve a better permeation of drugs through the blood–brain barrier. Lactose-appended β-CD successfully increased delivery of the model drug into the mouse brain [163]. They can also improve the absorption of curcumin and resveratrol, known for their neuroprotective properties in neurodegenerative diseases such as Huntington’s, Alzheimer’s and Parkinson’s diseases [164,165,166]. The addition of β-CD to curcumin β-CD nanoconjugates increased the water solubility of the drug and greatly reduced its toxicity with possible applications in preventing neurodegenerative diseases [167].

#### 4.1.2. Niemann–Pick Disease Type C (NPC)

NPC is a rare genetic disease determined by the accumulation of cholesterol and several other lipids in cells, including brain cells. This accumulation leads to severe and progressive neuronal degeneration.

HP-β-CD is approved as an orphan drug for treating NPC by FDA and EMA as it has shown a promising effect in preclinical trials. It removes accumulated cholesterol from neuronal cells, thereby facilitating its transport and elimination from the brain [168]. Preclinical studies showed no results after i.v. administration HP-β-CD to mice [169]. Thus, it was concluded that HP-β-CD cannot cross the blood–brain barrier which led to intrathecal administration, a more invasive route of administration. Intrathecal administration HP-β-CD to mice and cats with Niemann–Pick type C disease decreased the progression of neuronal damage [170,171]. Clinical studies have shown that HP-β-CD can slow the progression of Niemann–Pick type C disease in patients, reducing neurodegenerative symptoms and improving quality of life [172]. Furthermore, long-term intrathecal administration of HP-β-CD stabilized patients with ages ranging from 1.5 and 20 years. However, the outcome depends on the progression of the disease when diagnosis is given [173].

#### 4.1.3. Alzheimer’s Disease

In Alzheimer’s disease (AD), there is an abnormal build-up of proteins, particularly β-amyloid, which form plaques in the brain [174,175]. This leads to neuronal degeneration and cognitive loss. Furthermore, high cholesterol levels may constitute a potential risk in developing Alzheimer’s disease as it increases neuronal internalization of β-amyloid peptides [159,176]. Cyclodextrins may play a role in reducing these toxic accumulations.

Preclinical studies suggest that cyclodextrins could inhibit the accumulation of amyloid plaques by facilitating their removal or preventing their formation. In addition, they may improve the stability and solubility of drugs used to treat Alzheimer’s disease symptoms, such as acetylcholinesterase inhibitors [81]. γ-CD forms an inclusion complex with crocetin to deliver the drug across the BBB as a potential treatment for AD. Moreover, it also increased the solubility and bioavailability of the drug. The inclusion complex exhibited no toxicity towards normal neuronal cells [177]. Cyclodextrins increase water solubility and antioxidant effect of resveratrol and oxyresveratrol by forming inclusion complexes [178]. Oxyresveratrol-β-cyclodextrin determined an antioxidant effect and inhibited the activity of histone deacetylase in rats with AD. Furthermore, the inclusion complex reversed cognitive and behavioral deficit associated with the disease [179].

#### 4.1.4. Parkinson’s Disease

Parkinson’s disease involves the accumulation of α-synuclein, a protein that becomes toxic in excess, contributing to the death of dopaminergic neurons. Studies suggest that cyclodextrins could reduce the accumulation of α-synuclein, thereby protect neurons. Also, the use of cyclodextrins such as β-CD and HP-βCD in formulating drugs such as levodopa as controlled drug delivery systems can improve stability and bioavailability, reducing efficiency fluctuations and adverse effects [82,180,181]. Furthermore, a study showed that by administering HP-β-CD subcutaneously twice a week for two weeks to mice with induced Parkinson’s disease the loss of dopaminergic neurons was greatly reduced [182].

#### 4.1.5. A Reduction in Oxidative Stress and Neuroinflammation

Cyclodextrins can facilitate the delivery of antioxidants and anti-inflammatory drugs to the brain, thereby they help protect neurons against oxidative stress and inflammation, two important factors in the progression of neurodegenerative diseases. In Alzheimer’s and Parkinson’s diseases, oxidative stress is a major mechanism by which neurons are damaged [183]. Including antioxidants such as lipoic acid in cyclodextrins (γ-CD) can increase its stability and effectiveness at CNS level due to its ability to cross the BBB and produce an antioxidant and anti-inflammatory effect [184,185,186]. Inclusion complexes of lipoic acid and β-CD exhibiting increased the water solubility and photostability were also obtained in order to be administered orally [187].

### 4.2. Respiratory Diseases

Formulations with cyclodextrins are used to improve the stability and bioavailability of drugs administered by inhalation in the treatment of respiratory diseases, such as chronic obstructive pulmonary disease (COPD) and bacterial pneumonia [188].

Cyclodextrins are widely used in respiratory diseases, particularly due to their ability to improve drug formulations and reduce toxicity and adverse effects. The use of cyclodextrins in this field focuses on improving the bioavailability and stability of inhaled drugs, reducing the adverse effects of corticosteroids. Furthermore, they are used in the therapy of rare or chronic lung diseases such as cystic fibrosis and chronic obstructive pulmonary disease (COPD). Patent number 9034846B2 describes the use of HP-β-CD as an active component administered by inhalation in the treatment and prevention of asthma and COPD in mice. The formulation succeeded in significantly reducing levels of CXCL-1 compared to placebo group [188].

#### 4.2.1. Improving the Delivery of Inhalation Drugs

Cyclodextrins are frequently used as aerosol and powder formulations for inhalation because they can increase the solubility and stability of active substances. In respiratory diseases such as asthma and COPD, inhalation delivery is essential, and cyclodextrins help to deliver drugs efficiently to the airways. HP-β-CD, γ-CD and M-β-CD aqueous solutions can be used to increase the solubility of drugs in aerosol formulations in order to achieve pulmonary deposition [189].

#### 4.2.2. Reducing Corticosteroid Toxicity

Inhaled corticosteroids such as budesonide and fluticasone are the main drugs used in asthma and COPD. However, their long term use may produce local side effects such as irritation and upper respiratory tract infections. Cyclodextrin corticosteroid inclusion complexes reduce local irritation and excessive systemic absorption, which decreases the risk of side effects. Studies showed that inhaled corticosteroid formulations with cyclodextrins reduce the incidence of adverse effects such as oral thrush and dysphonia (hoarse voice), increasing the long-term safety of these treatments [81,190].

Budesonide, a corticosteroid used to treat asthma, has increased bioavailability when formulated with cyclodextrins. This improves pulmonary absorption, reducing the required dose and, subsequently, the risk of systemic adverse effects. A BUD:HP-β-CD inclusion complex displayed anticytotoxic and antioxidant properties due to HP-β-CD and anti-inflammatory properties due to budesonide [191]. Spray-dried microparticles containing a BUD:HP-β-CD inclusion complex were administered by inhalation in a mouse model of asthma, resulting in an increased anti-inflammatory effect [148]. This could lead to reduced systemic side effects that are normally a result of corticosteroid high doses [148].

An inclusion complex between hydroxypropyl-γ-cyclodextrin and fluticasone formulated as liposomes permitted a higher accumulation of fluticasone in the lungs compared to the drug alone [192].

#### 4.2.3. Cystic Fibrosis

In cystic fibrosis, the build-up of viscous mucus in the lungs leads to recurrent infections and severe breathing difficulties. Cyclodextrins are used to facilitate the delivery of inhaled antibiotics, such as tobramycin, directly to the lungs. Cyclodextrins help increase the solubility of antibiotics and improve their stability, which allows for a more effective action against chronic bacterial infections affecting patients with cystic fibrosis. Formulation of tobramycin with cyclodextrins improves lung distribution and reduces the frequency of administration, providing a more convenient and effective treatment [193]. Furthermore, Gunasekara showed that methyl-β-cyclodextrin could be used to restore airway surfactant function in patients with cystic fibrosis [194].

#### 4.2.4. Chronic Obstructive Pulmonary Disease (COPD)

COPD is characterized by chronic inflammation of the airways and airflow restriction. The main treatment includes bronchodilators and inhaled corticosteroids. The use of cyclodextrins in COPD drug formulations can increase the solubility of bronchodilators and anti-inflammatories, allowing a controlled and prolonged release of drugs [195].

Ipratropium and tiotropium, bronchodilators commonly used in COPD, are often formulated with cyclodextrins to increase the efficiency and duration of drug action in the airways, thereby reducing the number of daily doses required [195].

Salbutamol, a bronchodilator, was associated with two cyclodextrins, namely γ cyclodextrin and dimethyl-β-cyclodextrin (DMCD) in order to obtain inclusion complexes formulated as dry powder aerosols. Thus, the solubility and dissolution rate of the drug were increased. The release of the drug was fast (within 5 min) for both formulations and nearly complete over 30 min [196].

#### 4.2.5. Reducing the Irritating Effects of Inhaled Drugs

Certain inhaled drugs can produce irritation or local side effects in the respiratory tract. Cyclodextrins form inclusion complexes with these substances, preventing direct contact with the respiratory mucosa and thus reducing adverse effects. In the treatment of asthma, bronchodilators may cause rare paradoxical bronchospasm or irritation. Formulations with cyclodextrins help avoid these problems by providing gradual and localized drug release [197].

Cyclodextrins play a crucial role in the treatment of respiratory diseases by improving inhaled delivery of drugs, reducing toxicity of corticosteroids and facilitating controlled release of antibiotics and bronchodilators. These advantages not only increase the effectiveness of the treatment, but also reduce the risk of adverse effects, thus contributing to a safer and more effective therapy for patients with chronic and acute respiratory conditions.

### 4.3. Cancer Therapy

Cyclodextrins are integrated into oncology treatments to improve targeting and delivery of chemotherapeutic agents, such as doxorubicin. These formulations help increase solubility and selective delivery to tumor cells, enhancing treatment efficacy [198].

Cyclodextrins play an important role in cancer therapy due to their ability to improve the solubility, stability and bioavailability of anticancer drugs. They also help reduce the toxicity associated with chemotherapy, thus improving the effectiveness of the treatment and quality of life [198]. Some of the main ways cyclodextrins are used in cancer therapy include improved drug solubility, reduced toxicity, targeted therapy or gene transport.

#### 4.3.1. Improving Drug Solubility and Bioavailability

Many anticancer drugs, such as paclitaxel, doxorubicin or camptothecin, have low solubility in water, which limits their therapeutic efficiency. Cyclodextrins, especially β cyclodextrin and its derivatives (HP-β-CD, SBE-β-CD), may form inclusion complexes with these hydrophobic drugs, increasing their solubility and facilitating their intravenous or oral administration [199].

Paclitaxel, used to treat various types of cancer (breast, ovarian, lung), has very low water solubility, requiring toxic solvents that cause severe side effects. HP-β-CD is used to replace these solvents and improve the solubility of paclitaxel, reducing systemic toxicity [200]. Furthermore, liposomes formulated with PTX-HP-β-CD inclusion complex demonstrated increased inhibition of tumor growth [201].

#### 4.3.2. A Reduction in Toxicity Associated with Chemotherapy

Chemotherapy is often associated with high toxicity, affecting both cancer cells and healthy cells. Cyclodextrins can protect the body from these adverse reactions by selectively delivering drugs to the tumor site, thus reducing systemic toxicity. Including doxorubicin in cyclodextrins can reduce the cardiac toxicity associated with this anticancer drug, protecting the heart without diminishing the effectiveness of the treatment. Preclinical studies have shown that inclusion complexes with cyclodextrins protect healthy cells from doxorubicin damage [202].

An oral formulation containing an inclusion complex of satraplatin and β-cyclodextrin displayed increased bioavailability and antitumor effect. Moreover, intestinal mucosa was not affected after oral administration in xenograft mice model [203].

#### 4.3.3. Targeted Therapy

Cyclodextrins are used for the development of targeted delivery systems, which allow the drug to reach cancer cells directly, while minimizing the impact on healthy cells. A high therapeutic effect with fewer side effects is achieved through this mechanism.

In ovarian or breast cancers, cyclodextrins may be used in combination with nanoparticles to provide controlled and targeted delivery of drugs, namely tamoxifen [204] or etoposide, to the tumor, thereby reducing the required systemic dose [205].

Curcumin was encapsulated in crosslinked cyclodextrin nanoparticles in order to achieve targeted delivery. The nanoparticles extended curcumin antitumoral activity, resulting in up to 90% cell death in cervical cancer cells [206].

Cyclodextrins are also used to develop drug delivery systems designed for active targeting and stimuli-responsive nanomedicine in order to achieve increased drug concentrations at tumor sites. Hydroxycamptothecin was encapsulated in a multiple-component nanocomposite for targeting CD 44 receptors and the nucleus. The nanocomposites included carboxymethyl-β-cyclodextrin associated with protamine to increase nuclear localization and membrane penetration. The nanoparticles displayed good antitumor effect in vivo [207]. Liang et al. successfully developed a DOX-loaded nanocomposite achieving chemo-photothermal therapy of hepatoma cells and CD44 receptor active targeting. β-cyclodextrin-hyaluronic acid polymers were used to increase drug loading, while Fe_3_O_4_–graphene oxide was used to obtain a photothermal response mechanism [208].

#### 4.3.4. Bioactive Molecules Transport (Genes or RNA)

Cyclodextrins can be used in gene or RNA-based therapies (siRNA) to selectively deliver therapeutic molecules to cancer cells. These therapies are innovative in the treatment of cancer, but one of the major challenges is the effective delivery of these molecules at a cellular level. Cyclodextrins can form complexes with RNA or DNA. This protects them from degradation and facilitates penetration into tumor cells.

Preclinical studies have shown that cyclodextrins can be used to deliver siRNA that targets genes involved in cancer cell proliferation. These cyclodextrin-based delivery systems show promise as personalized cancer therapies [209].

Docetaxel and siRNA were formulated into folate-functionalized PEGylated CD-based nanoparticles to achieve a synergistic effect in inhibiting tumor growth [210].

Antibody targeted cyclodextrin-based nanoparticles were formulated for targeted delivery of siRNA to leukemia stem cells. The formulation exhibited greater therapeutic effect in relapsed acute myeloid leukemia (AML) samples compared to newly diagnosed AML [211].

#### 4.3.5. Reducing Drug Resistance

Cyclodextrins may play a role in overcoming cancer cell drug resistance, which is a common problem in cancer treatment. By forming complexes with anticancer drugs, cyclodextrins can prevent the efflux mechanisms or inactivation of the drug, thereby improving the effectiveness of treatment. In the case of doxorubicin treatments, some tumors develop resistance by rapidly expelling the drug from the cells. Formulation with cyclodextrins may help prevent this phenomenon, increasing the tumor sensitivity to chemotherapy. DOX was loaded into nanoparticles based on star-shaped polymers of β-CD and d-α-tocopherol polyethylene glycol succinate. The nanoparticles demonstrated a superior cytotoxic effect in drug-resistant cancer cells compared to DOX both in vitro and in vivo [212].

Cyclodextrins display several advantages in cancer therapy such as increased solubility and bioavailability, reducing chemotherapy toxicity and improving targeted delivery of the drug. They have shown an increased potential in the development of safer and more efficient therapies, by offering solutions to several challenges associated with cancer therapy.

### 4.4. Cardiovascular Diseases

Cyclodextrins are researched as a treatment for cholesterol related cardiovascular diseases, improving solubility and efficiency of lipid modifying agents. Furthermore, they are used to treat cardiovascular diseases due to their capacity to improve drug solubility, bioavailability and efficiency and also to reduce both side effects and toxicity associated with conventional therapy. In cardiovascular diseases, they may be used to optimize existing treatments and to address problems such as inflammation, lipid metabolism and cardiac toxicity [213].

#### 4.4.1. Reducing Cardiac Toxicity

Cyclodextrins can be used to reduce toxic cardiac effects of several drugs. Anticancer therapies such as doxorubicin are known for their severe cardiotoxicity. Cyclodextrins, especially modified ones such as HP-β-CD, can reduce lipid accumulation in cardiac cells, thereby protecting the heart from the toxic effects of these treatments. The use of cyclodextrins showed a significant reduction in cardiotoxicity associated with doxorubicin treatment, thus allowing the continuation of therapy without severe deterioration of cardiac function [214]. A study conducted by Durco et al. determined that d-limonene (DL) and HP-β-CD inclusion complex has a cardioprotective effect during treatment with DOX, protecting the heart against arrhythmias induced by DOX [215].

#### 4.4.2. Regulating Lipid Metabolism

Cyclodextrins play a key role in managing lipid metabolism, a critical component in cardiovascular diseases such as atherosclerosis. The accumulation of cholesterol in arterial walls leads to the formation of atheromatous plaques, which can cause heart attacks or strokes.

Cyclodextrins, especially HP-β-CD, have the ability to mobilize and facilitate cholesterol elimination from the macrophages in atherosclerotic plaques, thus helping to reduce the risk of cardiovascular disease [216]. Preclinical studies have shown that HP-β-CD can reduce cholesterol accumulation in atherosclerotic plaques and reduce atherosclerotic plaques in carotid and coronary arteries, thus preventing the progression of atherosclerotic disease. Furthermore, administration of HP-β-CD also lowered lipid levels and uric acid levels and improved liver function. This paves the way for the use of cyclodextrins in the prevention and treatment of cardiovascular disease [215,217].

#### 4.4.3. Formulation of Antihypertensive Medication

Cyclodextrins may improve the bioavailability of drugs used to treat hypertension and other cardiovascular disorders. Antihypertensive medication such as nifedipine or enalapril exhibits low solubility or stability and subsequently a low absorption. Formulating nifedipine in inclusion complexes increases its solubility and therapeutic efficiency [218]. Inclusion complexes of nifedipine were obtained with β-cyclodextrin and other cyclodextrin derivatives. The highest dissolution rate was observed for β-cyclodextrin followed by hydroxypropyl-β-cyclodextrin and DIMEB [219]. Nifedipine has an increased bioavailability when formulated into cyclodextrins which allows for a faster absorption [218]. Enalapril has a low stability at a pH level of 5 or above. Complexation with β-cyclodextrin increased stability with applications in solid dosage form formulation [220].

Hydrochlorothiazide (HCT) is an FDA approved diuretic for children. However, there are no commercial liquid formulations due to low water solubility and stability. HCT–HP-β-CD and HCT:SBE-β-CD inclusion complexes were included into solid lipid nanoparticles and formulated as a pediatric oral suspension. The new formulations exhibited enhanced bioavailability and diuretic effect [221].

#### 4.4.4. Therapy of Heart Failure

Heart failure is often associated with the accumulation of cholesterol and lipids in heart tissues, which exacerbates the patient’s condition. Cyclodextrins can be used to reduce this phenomenon by mobilizing cholesterol and facilitating its removal from cardiac cells. The use of cyclodextrins in animal models of heart failure has demonstrated a reduction in cholesterol accumulation in the heart, which may provide a new therapeutic approach for patients with heart failure. Carvedilol, a β-blocker administered orally for the treatment of chronic heart failure, exhibits rapid first pass metabolism and poor bioavailability [222]. HP-γ-CD and poly(vinyl pyrrolidone) were used to obtain a ternary complex with carvedilol which reduced crystallinity of the drug. The ternary complex was further developed as tip microarray patches. The microarray patches led to sustained plasma levels of carvedilol over a period of seven days in Sprague Dawley rats, making them a suitable alternative to oral administration treatment of chronic heart failure [223].

#### 4.4.5. Prevention of Cardiovascular Inflammation

Inflammation plays a key role in many cardiovascular diseases, including atherosclerosis and other inflammatory conditions of the heart and blood vessels. Cyclodextrins can help reduce inflammation by removing lipids that trigger inflammatory responses and by improving the delivery of anti-inflammatory drugs to the affected tissues. HP-β-CD has been used in studies to reduce inflammation associated with cholesterol build-up in arteries, thereby helping to prevent and control inflammation-related cardiovascular diseases [119,224]. Furthermore, HP-β-CD could be used to treat chronic inflammation after stroke and prevent post stroke dementia [225].

### 4.5. Infectious Diseases

3Cyclodextrins have started to gain significant importance in the treatment of infectious diseases due to their ability to improve the solubility, bioavailability and antimicrobial effect [139,226,227]. They modify the release profile of antibiotics and antifungals and decrease drug degradation while reducing associated side effects [133].

#### 4.5.1. Improving the Solubility of Antibiotics

Many antibiotics have low solubility in water, which limits their bioavailability and therapeutic effectiveness. Cyclodextrins may form inclusion complexes with these drugs, improving their solubility and thus their effectiveness. A novel HP-β-CD polymer was used to improve the solubility of azithromycin, commonly used to treat respiratory infections. This formulation helps improve the release of the drug and reduce the doses required [228]. Complexation of clarithromycin with β-CD increased the solubility of the drug by 700 fold at pH 7.4 [229]. Inclusion complexes of norfloxacin and both β-CD and HP-β-CD prepared by the freeze-drying method exhibited a higher dissolution rate and better solubility [230].

#### 4.5.2. Side Effects Reduction

Cyclodextrins may help minimize side effects associated with antibiotics. They form inclusion complexes with antibiotics and protect healthy cells from the toxicity generated by broad-spectrum antibiotics.

Associating HP-β-CD with gentamicin showed a decrease in renal toxicity which allows an effective treatment without compromising the renal function. Inclusion complexes of β-CD with chloramphenicol and N-acetylcysteine were prepared to improve solubility and inhibit biofilm formation. Furthermore, the complexes reduced antibiotic toxicity against leukocytes [231]. Amikacin microspheres were obtained by encapsulation into β-CD in order to reduce drug toxicity and improve its efficacy [232].

#### 4.5.3. Targeted Drug Delivery

Cyclodextrins can be used to develop targeted drug delivery systems that allow the drug to reach the infection site directly. This is particularly important to treat severe infections or pathogens that are resistant to antibiotics. In the treatment of bacterial infections, cyclodextrins are used in combination with nanoparticles to deliver antibiotics directly into infected cells, increasing the effectiveness of the treatment.

SBE-β-CD was used to increase encapsulation efficiency and loading of levofloxacin into chitosan nanoparticles. The nanoparticles displayed a sustained release of the antibiotic over 72 h. Furthermore in vitro antibacterial studies demonstrated a doubled activity against Gram-positive and Gram-negative bacteria compared to the free drug [233]. Inclusion complexes of roxithromycin and β-CD and HP-β-CD were encapsulated into PLGA nanoparticles. The nanoparticles exhibited significant antibacterial activity against multidrug-resistant Gram-positive and Gram-negative bacteria [234].

#### 4.5.4. Treatment of Fungal Infections

Cyclodextrins are also used in the formulation of antifungals such as amphotericin B, which is known for its toxicity. Cyclodextrins help reduce the side effects of this drug by improving solubility and bioavailability. Formulations of amphotericin B with cyclodextrins have shown increased efficacy in the treatment of systemic fungal infections while reducing the toxicity associated with administration of this drug. Sertaconazole is an antifungal agent used to treat *Candida albicans* infections, with poor water solubility. Sertaconazole complexed with HP-β-CD was formulated as hydrogels which displayed a controlled release of the drug for more than 24 h [235].

#### 4.5.5. Combating Drug Resistance

Cyclodextrins can help overcome antibiotic resistance, which is a major concern when treating bacterial infections. They can help maximize therapeutic effect by ensuring efficient delivery of antibiotics. In infections caused by drug-resistant bacteria, the use of cyclodextrins can improve the effectiveness of drugs, thus allowing more effective treatments for difficult-to-treat infections. Antibiotic–cyclodextrin inclusion complexes allow the administration of lower drug concentrations, which delays antibiotic resistance [236].

Methicillin was complexed with a designed β-CD derivative in order to increase its antibacterial potency against two methicillin-resistant Staphylococcus aureus strains, resulting in an antibacterial activity similar to anti-MRSA antibiotics such as linezolid and vancomycin [237].

## 5. Future Perspectives

The field of cyclodextrin-based drug delivery systems continues to evolve, with ongoing research exploring new applications, modifications, and formulations. Future advancements are expected as the development of tailored drug delivery systems incorporating cyclodextrins will allow for more precise and individualized treatments. Advances in nanotechnology and molecular biology will enable the design of cyclodextrin-based carriers that selectively target diseased tissues, reducing systemic side effects and enhancing therapeutic outcomes. Researchers are also investigating cyclodextrin-based formulations that respond to specific stimuli such as pH, temperature, light, or enzymatic activity. These smart drug delivery systems can provide controlled and on-demand drug release, improving treatment efficacy and patient compliance [50,51,207,208].

Cyclodextrins show great promise in facilitating the delivery of gene therapies, including siRNA and mRNA, by improving stability and cellular uptake [209,210,211]. Their potential role in non-viral gene delivery systems could revolutionize treatments for genetic disorders, cancers, and viral infections.

Beyond their established use in solubility enhancement and controlled release, cyclodextrins are being explored for applications in neurodegenerative diseases [172,173,174,175,176], cardiovascular conditions [220,221,238], and respiratory disorders [192,193,194]. Their ability to interact with lipophilic molecules makes them promising candidates for cholesterol regulation and toxin removal in various diseases. Despite significant advancements, the number of commercially available cyclodextrin-based pharmaceutical products remains limited [238]. Future efforts should focus on clinical applications by addressing safety concerns, optimizing large-scale production, and securing regulatory approvals for new formulations.

## **6.** Conclusions

Cyclodextrins have revolutionized drug delivery by enhancing the solubility, bioavailability, and therapeutic efficacy of numerous active pharmaceutical ingredients. Their multifunctional role in pharmaceutical formulations—ranging from stabilizing drugs to enabling targeted and controlled release—demonstrates their immense potential in modern medicine. While cyclodextrins have already improved treatment strategies for a variety of diseases, continued research will be essential to unlock their full capabilities. Innovations in nanotechnology, bioconjugation, and hybrid drug delivery systems will further expand their applications in precision medicine and complex disease management. In the coming years, interdisciplinary collaboration between pharmaceutical scientists, biomedical engineers, and clinicians will drive the development of next-generation cyclodextrin-based therapies. With sustained research and commercial interest, cyclodextrins will continue to play a critical role in improving drug formulation strategies, ultimately leading to safer, more effective, and patient-friendly treatments.

## Figures and Tables

**Figure 1 pharmaceutics-17-00288-f001:**
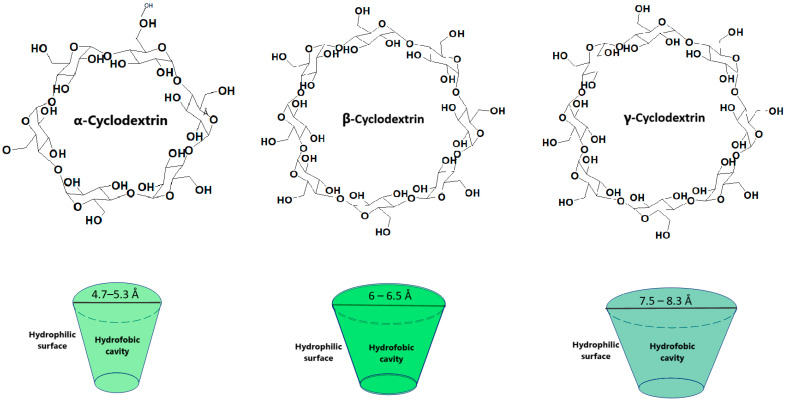
Cyclodextrin structure and inclusion complex.

**Figure 2 pharmaceutics-17-00288-f002:**
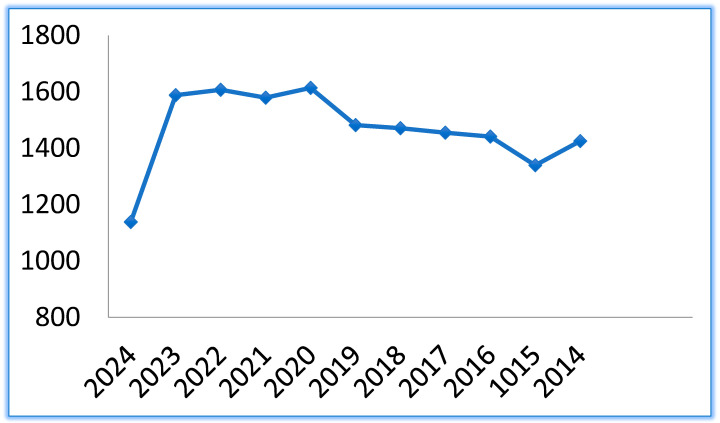
Graph of publications related to cyclodextrins in the last 10 years.

**Figure 3 pharmaceutics-17-00288-f003:**
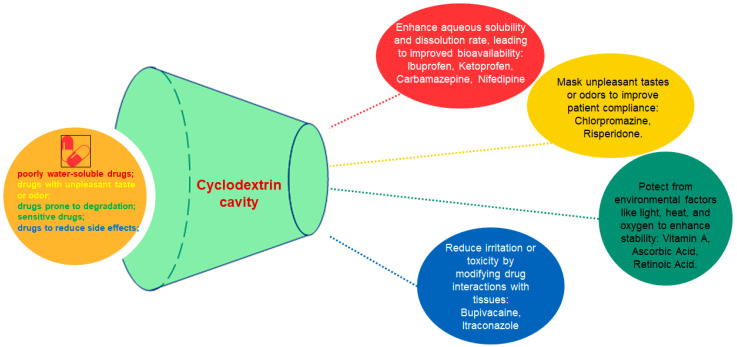
The effect of cyclodextrins over the encapsulated drug.

**Figure 4 pharmaceutics-17-00288-f004:**
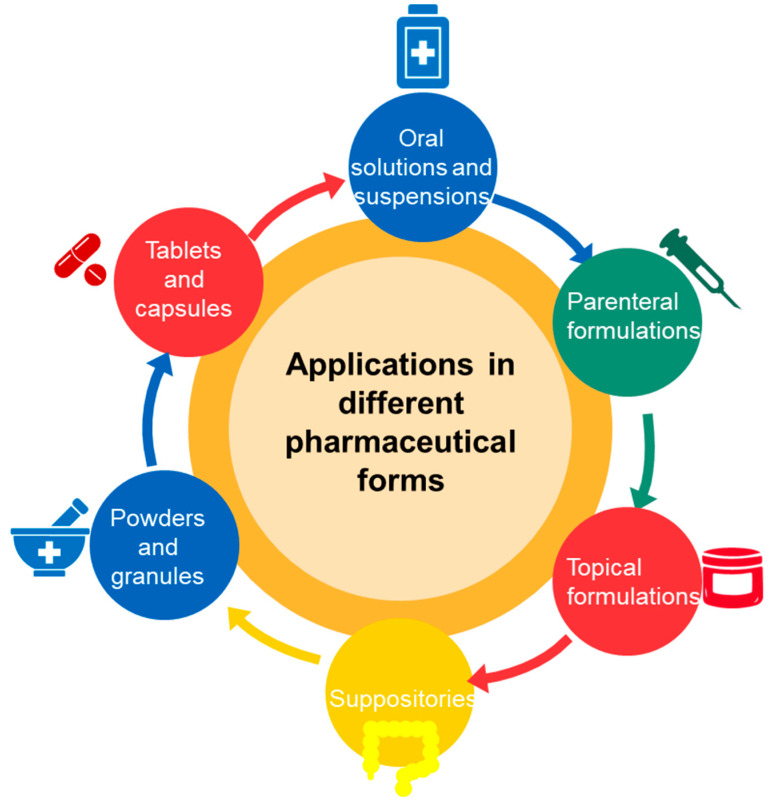
Pharmaceutical forms utilizing cyclodextrins for enhanced drug delivery.

**Table 1 pharmaceutics-17-00288-t001:** Cyclodextrins enhance the solubility of several active substances.

Active Substance	Water Solubility (mg/mL)	Solubility When Associated with β-Cyclodextrin (mg/mL)	Cyclodextrin Used	Reference
Amphotericin B	0.001	0.15	β-cyclodextrin sulfobutyl ether (SBE-β-CD)	[14]
Cefixime	0.76	17.5	SBE-β-CD	[15]
Cefpodoxime Proxetil	0.241	0.637	β-CD	[16]
Ceftiofur	0.03	2.18	HP-β-CD	[17]
Dexamethasone	0.1	2.5	β-CD	[18]
Diclofenac	4.0	20.0	HP-β-CD	[19]
Ibuprofen	0.1	10.0	Methyl-β-cyclodextrin (M-β-CD)	[20]
ITH12674	0.31	10.7	HP-β-CD	[7]
Itraconazole	0.001	4–5	HP-β-CD	[21]
Metronidazole benzoate	0.14	1.39	β-CD	[22]
Nifedipine	0.02	1.5	β-CD	[23]
Paclitaxel	0.003	2.0	HP-β-CD	[24]
Tinidazole	3.76	36.89	β-CD	[25]

**Table 2 pharmaceutics-17-00288-t002:** Cyclodextrins in tablets: enhancing drug performance and stability.

Active Substance	Trade Name and Formulation	Cyclodextrin	Effect of Cyclodextrin over the Active Substance	Indication	Reference
Brivaracetam	Brivlera	β-CD	Improved drug dissolution	Epilepsy	[71,72]
Cetirizine	Zyrtec, WalZyr, Revicet One-day tablets, chewing tablets	β-CD	Higher stability against oxidative degradation of the drug, increased solubility, masking of the unpleasant bitter taste of the drug	Allergy	[66,67,73,74]
Desloratadine	Desloratadine	β-CD	Increased stability (no coloration), minimum N-, formyl desloratadine formation	Allergy	[75]
Ethinylestradiol/ Drospirenone	Yaz^®^	β-CD	Enhance water solubility	Contraception	[64,76]
Famotidine	Famotidine OD Tablets	β-CD	Increased water solubility, mask bitter taste	Ulcer	[77]
Nimesulide	Nimedex	β-CD	Increased water solubility and dissolution rate, leading to rapid absorption of the drug, more rapid onset of action compared to nimesulide	Pain after arthroscopic surgery	[78,79]
Nitroglycerin	Nitropen, sublingual tablet	β-CD	Low volatility of nitroglycerine	Angina pectoris	[68,80]
Limaprost alfadex (Prostaglandin E1 analogue)	Opalmon tablets	β-CD	Increased stability against humidity	Pain, ulcer	[66,81,82]
Dinoprostone (Prostaglandin E2 derivative)	Prostarmon E (sublingual tablets), Dinoproston- betadex	β-CD	Increased stability, improved drug dissolution in saliva and organoleptic properties	Induction of labor	[66,81]
Rofecoxib	Rofizgel	β-CD	Improved solubility and dissolution rate	Arthritis	[66,83,84]
Tiaprofenic acid	Surgamyl	β-CD	Increase in solubility, absorption and bioavailability, masking bitter taste	Pain management	[85]

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
