# Peer review of "Cyclodextrins: Enhancing Drug Delivery, Solubility and Bioavailability for Modern Therapeutics"

_pharmaceutics, 2025, doi:10.3390/pharmaceutics17030288_

Round 1

Reviewer 1 Report

Comments and Suggestions for Authors

Remove the period from the title of Table 1.

Check the spelling and grammar of the information presented in Tables 5 and 7.

Author Response

Dear Reviewer,

We would like to thank you for your accurate observations and valuable comments. We used all these and improved the paper accordingly. We have integrated in the manuscript the answers to your questions.

All changes in the revised manuscript were marked up using the “Track Changes” function.

Thank you once again for your constructive comments.

  1. Remove the period from the title of Table 1.

Answer: We removed the period.

  1. Check the spelling and grammar of the information presented in Tables 5 and 7.

Answer: We reviewed both Table 5 and 7.

Kind regards,

Mocanu Andreea Gabriela, PhD

Reviewer 2 Report

Comments and Suggestions for Authors

The manuscript pharmaceutics-3478120 “Cyclodextrins: Enhancing Drug Delivery, Solubility and Bioavailability for Modern Therapeutics” by Oana Elena Nicolaescu et al. reviews the use of cyclodextrins for the development of innovative dosage forms with improved pharmacological activity and reduced safety profile for various medical applications.

The topic of review is relevant; Nevertheless, the manuscript seems unfinished. The article requires major revision before acceptance for publication.

Comments and questions:

1)  The Introduction should be rewritten. The introduction should describe the importance and relevance of the topic, as well as provide examples of publications that have addressed the topic and highlight the novelty of your review. 

2) The title of Section 2 “Mechanism of action of cyclodextrins over the encapsulated active substance” does not correspond to its content, as the mechanisms of changes in the properties of drugs are not presented in this Section, except for the mechanism of solubility enhancement. The titles of Section 2 and Figure 3 should be changed.

3) Section on the use of cyclodextrins for the development of nanotechnology-based drug delivery systems should be added.

4)  Conclusion should be added to summarize the advantages and disadvantages of using cyclodextrins for biomedical applications.

5) The Future Perspectives Section should be added; and further development of the topic should be discussed. What promising directions for the use of cyclodextrins for pharmaceutical development of innovative dosage forms do you suggest?

Author Response

Dear Reviewer,

We would like to thank you for your accurate observations and valuable comments. We used all these and improved the paper accordingly. We have integrated in the manuscript the answers to your questions.

All changes in the revised manuscript were marked up using the “Track Changes” function.

Thank you once again for your constructive comments.

  • The Introduction should be rewritten. The introduction should describe the importance and relevance of the topic, as well as provide examples of publications that have addressed the topic and highlight the novelty of your review. 

Answer: We expanded the introduction.

2) The title of Section 2 “Mechanism of action of cyclodextrins over the encapsulated active substance” does not correspond to its content, as the mechanisms of changes in the properties of drugs are not presented in this Section, except for the mechanism of solubility enhancement. The titles of Section 2 and Figure 3 should be changed.

Answer: We changed the titles of both section 2 and figure 3.

  • Section on the use of cyclodextrins for the development of nanotechnology-based drug delivery systems should be added.

Answer: We added a section for nanotechnology-based drug delivery systems.

  • Conclusion should be added to summarize the advantages and disadvantages of using cyclodextrins for biomedical applications.

Answer: We added the conclusion section.

5) The Future Perspectives Section should be added; and further development of the topic should be discussed. What promising directions for the use of cyclodextrins for pharmaceutical development of innovative dosage forms do you suggest?

Answer: We added a future perspectives section.

Kind regards,

Mocanu Andreea Gabriela, PhD

Reviewer 3 Report

Comments and Suggestions for Authors

The authors present a comprehensive review dedicated on the recent achievements in field of cyclodextrin-based (nano)systems for drug-delivery applications. Special attention has been paid to various cyclodextrin-containing formulations (tablets, powders, capsules, parenteral and topical formulations etc.) and their application for treating different diseases. More specifically, the potential application of cyclodextrin-based drug delivery systems for treating cancer, neurodegenerative disordersrespiratory, cardiovascular and infectious diseases are discussed. The manuscript is well-organized, well-written and could be of interest for the readers of Pharmaceutics working in the field of drug delivery. I have the following remarks and suggestions:

1.     The advantages of the presence of hydroxyl functional groups on the surface of the cyclodextrins should be outlined (maybe in the introduction section). These functionalities allow further modifications, attachment of targeting ligands etc. and should not be underestimated.

2.     Page 7 (2.3.2 Controlled release of the drug). The authors explain the formation of the inclusion complex between the active compound and the cyclodextrin cavity as well as the advantages of the controlled drug release. However, the mechanism of the drug release from the cavities is not discussed at all. I think it would be useful for the readers if authors explain briefly in this subsection of the manuscript what is the driving force of the controlled drug release in the target sites and not during the systemic circulation, for example.

3.     Tables 2, 5, 6 and 7 are not mentioned in the text of the discussion.

4.     Page 18 (lines 491-500). It might be clearer for the readers if the authors mention that amino functionalized cyclodextrins are researched as delivery vehicles for gene therapy.

5.     Pages 23 and 26 (Targeted drug delivery): The authors give some examples for targeted drug delivery using cyclodextrin-based systems. However, examples of active targeting via attached to the systems’ surface ligands are missing. There are a number of references describing cyclodextin-based drug delivery systems designed for active targeting. This is just one recent example: Qin et al, Dual-targeted and esterase-responsive cyclodextrin-based host-guest nanocomposites for enhanced antitumor therapy. Colloids and Surfaces B: Biointerfaces, 246, 114371, 2025 (https://doi.org/10.1016/j.colsurfb.2024.114371). I think the authors could add a couple of references (not exactly the above-mentioned one) dealing with cyclodextin-based drug delivery systems for active ligand-mediated targeting.

Author Response

Dear Reviewer,

We would like to thank you for your accurate observations and valuable comments. We used all these and improved the paper accordingly. We have integrated in the manuscript the answers to your questions.

All changes in the revised manuscript were marked up using the “Track Changes” function.

Thank you once again for your constructive comments.

The authors present a comprehensive review dedicated on the recent achievements in field of cyclodextrin-based (nano) systems for drug-delivery applications. Special attention has been paid to various cyclodextrin-containing formulations (tablets, powders, capsules, parenteral and topical formulations etc.) and their application for treating different diseases. More specifically, the potential application of cyclodextrin-based drug delivery systems for treating cancer, neurodegenerative disorders, respiratory, cardiovascular and infectious diseases are discussed. The manuscript is well-organized, well-written and could be of interest for the readers of Pharmaceutics working in the field of drug delivery. I have the following remarks and suggestions:

  1. The advantages of the presence of hydroxyl functional groups on the surface of the cyclodextrins should be outlined (maybe in the introduction section). These functionalities allow further modifications, attachment of targeting ligands etc. and should not be underestimated.

Answer: We added the advantages of the hydroxyl functional groups in the controlled release section.

  1. Page 7 (2.3.2 Controlled release of the drug). The authors explain the formation of the inclusion complex between the active compound and the cyclodextrin cavity as well as the advantages of the controlled drug release. However, the mechanism of the drug release from the cavities is not discussed at all. I think it would be useful for the readers if authors explain briefly in this subsection of the manuscript what is the driving force of the controlled drug release in the target sites and not during the systemic circulation, for example.

Answer: we explained the mechanism of the drug release.

  1. Tables 2, 5, 6 and 7 are not mentioned in the text of the discussion.

Answer: We mentioned the tables in the text.

  1. Page 18 (lines 491-500). It might be clearer for the readers if the authors mention that amino functionalized cyclodextrins are researched as delivery vehicles for gene therapy.

Answer: We adjusted the text between lines 491 and 500.

  1. Pages 23 and 26 (Targeted drug delivery): The authors give some examples for targeted drug delivery using cyclodextrin-based systems. However, examples of active targeting via attached to the systems’ surface ligands are missing. There are a number of references describing cyclodextin-based drug delivery systems designed for active targeting. This is just one recent example: Qin et al, Dual-targeted and esterase-responsive cyclodextrin-based host-guest nanocomposites for enhanced antitumor therapy. Colloids and Surfaces B: Biointerfaces, 246, 114371, 2025 (https://doi.org/10.1016/j.colsurfb.2024.114371). I think the authors could add a couple of references (not exactly the above-mentioned one) dealing with cyclodextin-based drug delivery systems for active ligand-mediated targeting.

Answer:  We added a couple of studies describing cyclodextin-based drug delivery systems for active ligand-mediated targeting.

Kind regards,

Mocanu Andreea Gabriela, PhD

Round 2

Reviewer 2 Report

Comments and Suggestions for Authors

The manuscript may be accepted.